# Deep Hashing Based on the von Mises-Fisher Distribution for Fast Large-Scale Image Retrieval

## Abstract

Deep hashing has become a pivotal technology in image retrieval, benefiting from advancements in deep learning and the computational advantages of hashing methods. Existing methods either rely on Euclidean distance, ignoring directional similarity and leading to retrieval errors in high-dimensional scenarios, or use deterministic spherical projections that neglect feature probability distributions, making them susceptible to noise interference.To address these challenges, we propose von Mises-Fisher Deep Hashing (vMF-DH), which introduces the von Mises-Fisher (vMF) distribution to map features onto the unit hypersphere. This approach models directional distributions (cosine similarity) instead of relying on Euclidean distances, leveraging the maximum entropy property of the vMF distribution to enhance both adaptability and robustness. Additionally, we design the vMF-Hash loss function, which regulates feature clustering through the vMF concentration parameter. This ensures the generation of binary hash codes that are compact within classes, well-separated across classes, and highly discriminative. Extensive results on multiple benchmark datasets show that vMF-DH outperforms current state-of-the-art deep hashing methods, demonstrating superior performance in terms of retrieval accuracy and robustness.

## 1 Introduction

With the rapid advancement of image fast retrieval technology, hashing methods significantly reduce data storage costs by mapping high-dimensional image features to low-dimensional binary hash codes, enabling fast similarity retrieval through Hamming distance. Owing to their distinct advantages in storage efficiency and computational speed, hashing methods are widely applied in visual similarity searchWang et al. (2023a), clustering learningYoun et al. (2018), and information securityDe Guzman et al. (2019)Deepakumara et al. (2001).

Hash learning is typically classified into supervised and unsupervised categories, depending on whether semantic labels are utilized during the training process. Unsupervised hashing, which does not require manual annotation, is well-suited to scenarios with sparse labels, offering low cost and strong generalization capabilities. However, the hash codes generated by unsupervised methods tend to have limited discriminative power, which makes them less effective for high-semantic granularity retrieval tasks. In contrast, supervised hashing leverages label guidance to optimize distances in the Hamming space, resulting in hash codes with high semantic discriminative power. For specific datasets, supervised methods significantly outperform unsupervised approaches in terms of image retrieval accuracy Qin et al. (2023). Deep supervised hashing combines deep learning with hashing techniques, utilizing convolutional neural networks to link semantic labels with deep feature learning, thereby enhancing retrieval accuracy. However, traditional Euclidean distance methods are limited in modeling feature distributions. These methods assume that features follow Gaussian or uniform distributions, which makes it challenging to capture high-dimensional visual geometric features Zhang et al. (2022). Moreover, Hamming distance may fail to accurately assess sample similarity due to its sensitivity to symbol binarization, while direction-aware hashing overlooks spherical space features, rendering it vulnerable to background noise interference.

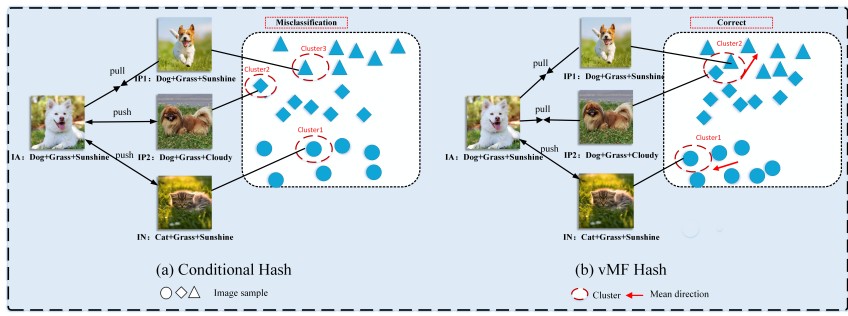

Figure 1: Traditional distance-based hashing methods (Fig. 1-a) are prone to classification errors when handling image clustering tasks. Consider samples $I_{P1}$ and $I_{P2}$, which exhibit noise differences only in their background labels. However, traditional methods fail to effectively capture feature directionality and other contextual information. Given the large distance values between these two samples, relying solely on distance information makes it difficult to correctly assign them to the same category.

Figure 1 illustrates the motivation behind our proposed framework. As shown in Figure 1-(a), $I_A$ and $I_{P1}$ (sharing "dog + grass + sunny sky"), along with $I_{P2}$ (sharing core elements "dog + grass" but differing in background noise), should exhibit similarity. However, traditional Euclidean hashing misjudges this due to symbol binarization and Hamming distance, overlooking the fact that high-dimensional feature similarity depends on direction rather than Euclidean radial distances. In contrast, as depicted in Figure 1-(b) (vMF Hash), our framework models the spherical distribution of high-dimensional features using a vMF distribution. By focusing on directional similarity, it captures the directional consistency of $I_A$, $I_{P1}$, and $I_{P2}$ (core label: Dog+Grass) for grouped classification while effectively separating irrelevant sample $I_N$. The binarization of hyperspherical features via the sign function maximizes the retention of directional semantic information, reducing the loss associated with traditional Euclidean binarization and improving retrieval performance.

To address these challenges, we propose the von Mises-Fisher Deep Hashing (vMFDH) framework, as illustrated in Figure 2. The core of this framework consists of two key components: (1) the vMF distribution model, which treats embedding vectors as mean directions and employs a concentration parameter to characterize the centrality of the distribution, thereby enabling precise modeling of the geometric structure of the embedding space; and (2) the loss function design, which adjusts the decision boundaries using a temperature parameter to maximize the probability of correct classifications while minimizing the impact of incorrect ones, thus guiding the model to learn more discriminative embedding representations. In summary, the main contributions of this paper are:

(1) We propose a hash learning framework incorporating the von Mises-Fisher (vMF) distribution, modeling feature vectors' directional distribution (cosine similarity) for more accurate semantic matching. Using vMF's hyperspherical maximum entropy property, the method better adapts to high-dimensional features, enhancing adaptability and generalization.

(2) We design a loss function based on the vMF distribution. This function preserves feature directional information while regulating feature clustering through the vMF concentration parameter, guiding the model to generate binary hash codes that are compact within classes, well-separated between classes, and highly discriminative.

(3) We conducted extensive experiments on image hashing retrieval across three benchmark datasets: MIRFLICKR-25K, NUS-WIDE, MS-COCO. Results show our proposed method outperforms traditional hashing approaches, validating the effectiveness of the vMF distribution in capturing image feature distributions and improving hashing retrieval accuracy.

## 2 Related Works

### 2.1 Supervised Hashing

Traditional supervised hashing methods such as Kernel-based Supervised Hashing Liu et al. (2012) (KSH) and Supervised Discrete Hashing Shen et al. (2015) (SDH) rely on manually designed features.To address this issue, Xia et al. proposed the first deep supervised hashing framework based on convolutional neural networks, Convolutional Neural Network Hashing Xia et al. (2014) (CNNH), which extracts continuous features in Euclidean space via convolutional neural networks. Liu et al. integrated feature learning with hash encoding, enabling convolutional neural networks to directly learn semantically relevant binary encodings. The proposed Deep Supervised HashingLiu et al. (2016) (DSH) framework achieves end-to-end training for deep supervised hashing. Cao et al. proposed the HashNet model Cao et al. (2017) by replacing the sign function with the tanh function, thereby circumventing the non-differentiability issue caused by the sign function. High-dimensional data inherently exhibits spherical distribution characteristics, rendering Euclidean distance-based methods ineffective at capturing directional semantic similarity. Yuan et al. proposed the CSQ frameworkYuan et al. (2020), which maps category labels to hash centers. This framework optimizes hash codes by maximizing the cosine similarity between network outputs and their corresponding centers. However, the constructed hash centers cannot guarantee minimum distance in worst-case scenarios . To address this, Wang et al. proposed an optimization scheme that solves for hash centers under the constraint of minimizing the distance between any two hash centers, namely the minimum-distance separated deep hash Wang et al. (2023b). Lu et al.'s Self-Paced Relational Contrastive Hashing (SPRCH) method balances synergistic effects between point-to-point and point-to-class relationshipsLu et al. (2024).

### 2.2 Unsupervised Hashing

Unsupervised hashing algorithms learn binary hash codes through predefined semantic distance metrics, making them suitable when data annotation is costly or difficult to obtain. Iterative Quantization (ITQ) optimized quantization error post-linear projection through alternating minimization Gong et al. (2013). Lin et al. proposed DeepBit,this framework minimizes quantization loss, enforces uniform hash code distribution, and adds rotation-invariant loss for robustness Lin et al. (2016). Yang et al. developed Semantic Structure-based Unsupervised Deep Hashing (SSDH) Yang et al. (2018), using pre-trained CNNs to extract deep features. However, it relies on static distance thresholds for pre-trained features, struggling to adapt dynamically to data distribution changes. To fix this, Shen et al. proposed an Unsupervised Deep Hashing with Similarity-Adaptive and Discrete OptimizationShen et al. (2018), which enables real-time adjustment of the similarity structure during the hash code optimization process.Qin et al. learned compact binary codes by fusing global and spatial representations, naming their method Unsupervised Deep Multi-Similarity Hashing with Semantic Structure (UDMSH) Qin et al. (2021). Dong et al. proposed Deep K-Means Hashing (DKMH) by minimizing pairwise supervised loss and optimizing binary K-means to generate discriminative hash codes Dong et al. (2021). Luo et al. proposed HashSIM, which remedies existing methods' deficiencies (similarity signal confidence discrepancy; insufficient hash code independence/robustness) via structural and intrinsic similarity learning Luo et al. (2022). To solve unreliable pseudo-labels and cluster number sensitivity, Meng et al. proposed UDHPM, optimized via a dynamic pseudo-multilabel generation network and KL divergence-based category information preservation strategy Meng et al. (2024).

### 2.3 vMF Distribution

The vMF distribution, as the maximum entropy distribution on hyperspheres, naturally adapts to the directional clustering properties of high-dimensional features and has been applied to tasks across multiple domains. For instance, in 2004, Banerjee et al. constructed a generative mixture model based on the von Mises-Fisher (vMF) distribution for clustering directional data on the unit hypersphere Banerjee & Ghosh (2004). In 2021, Scott et al. proposed a von Mises-Fisher (vMF) distribution loss function for embedding geometric properties in supervised learning,they demonstrated that the spherical loss Scott et al.

(2021). Recently, Zeng et al. extended the vMF distribution to construct a more adaptable distribution-guided geometric stochastic model for UAV-UAV geometric-based stochastic modeling (GBSM)Zeng et al. (2025).

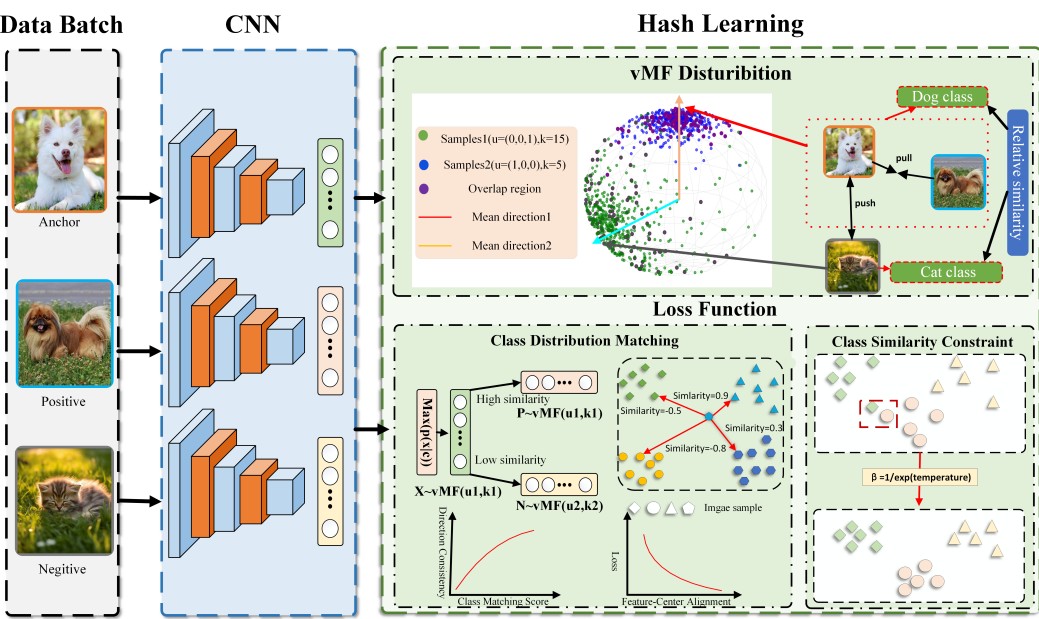

Figure 2: The proposed vMFDH framework primarily comprises two components: (1) vMF Distribution: Transforms embedding vectors into vMF distribution parameters via vMF projection, employing feature unitization direction $\boldsymbol{\mu}$ and L2 norm $\boldsymbol{\kappa}$ to precisely model high-dimensional spherical features; (2) Loss Function Module: A vMF-based log-likelihood function generates binary codes that are compact within classes and discriminative between classes.

## 3 Methodology

This section aims to elaborate on the vMFDH framework from two core perspectives: the von Mises-Fisher (vMF) distribution probabilistic classifier and the design of loss function.

### 3.1 vMF Distribution Probability Classifier

The von Mises-Fisher (vMF) distribution-based probabilistic classifier is a statistical learning model tailored for directional data. Its core mechanism involves probabilistically modeling feature vectors in high-dimensional unit spherical spaces using the vMF distribution, combined with Bayesian decision theory to perform classification. This approach overcomes the limitations of Euclidean space distributions like the Gaussian distribution when modeling spherical data, making it particularly well-suited for scenarios where features exhibit directional dominance.

Model training requires estimating two types of parameters from labeled data: the vMF distribution parameters for each class $(\boldsymbol{u}_i, k_i)$, and the class prior probability $p(c)$. During classification, the feature vector $\boldsymbol{x}$ of a new sample is first normalized to a unit vector. The likelihood probability $p(\boldsymbol{x}|c)$ is then computed based on the vMF parameters trained for each class. The posterior probability $p(c|\boldsymbol{x})$ is then computed using Bayes' theorem: $p(c|\boldsymbol{x}) \propto p(\boldsymbol{x}|c) \cdot p(c)$. Finally, $\boldsymbol{x}$ is assigned to the optimal category based on the Maximum A Posteriori (MAP) principle.

## 3.2 Loss Function

The vMF distribution is a classical probabilistic model for directional data on a sphere, suitable for describing the distribution characteristics of high-dimensional embedded vectors. The pdf for a d-dimensional unit vector x is:

$$p(\mathbf{x} \mid \boldsymbol{\mu}, \kappa) = \frac{\kappa^{\frac{d}{2}-1}}{(2\pi)^{\frac{d}{2}} I_{\frac{d}{2}-1}(\kappa)} \exp\left(\kappa\boldsymbol{\mu}^\top \mathbf{x}\right) \tag{1}$$

where $\boldsymbol{x}, \boldsymbol{\mu} \in \mathbb{S}^{n-1}$ and $\kappa \geq 0$, and $I_v$ denotes the modified Bessel function of the first kind of order $v$. Traditional classification losses (such as Softmax) only optimize "class discriminability" without constraining the alignment between feature distributions and spherical priors. By incorporating the vMF distribution, we ensure embedding vectors follow a spherical distribution while enhancing inter-class separability. For samples of class $y$, their embedding vectors $z$ follow vMF$(\boldsymbol{\mu}_y, \kappa_s)$ (where $\boldsymbol{\mu}_y$ is the "center direction" of class $y$ and $\kappa_s$ controls distribution density). During classification, we maximize the "log-likelihood" of $z$ for the true class $y$. Inspired by the Softmax loss, we construct an expected log-likelihood form:

$$\mathcal{L} = -\mathbb{E}_{z\sim\text{vMF}(\mu_y, \kappa_s)} \left[\log \frac{\exp\left(\beta\mu_y^\top z\right)}{\sum_{j=1}^{C} \exp\left(\beta\mu_j^\top z\right)}\right] \tag{2}$$

where $\beta > 0$, $C$ is the total number of classes, and $\boldsymbol{\mu}_j$ is the center direction of class $j$. During actual training, the model processes batch samples (batch size $= B$) and requires explicit parameterization of the "center" and "concentration" of the vMF distribution. Assume the true class of the $i$th sample is $y_i$, and its embedding vector $z_i$ follows vMF$(\boldsymbol{\mu}_{y_i}, \kappa_i)$ (where $\kappa_i$ is learnable). Simultaneously, the category center $\boldsymbol{\mu}_j$ is associated with the model weight $\boldsymbol{w}_j$ via $\boldsymbol{\mu}_j = \frac{\boldsymbol{w}_j}{\|\boldsymbol{w}_j\|}$. To refine the loss function further, introduce the logarithm of the vMF distribution normalization constant (derived from $C_d(\kappa)$ in equation (2)), yielding the final loss expansion:

$$\mathcal{L} = \frac{1}{B}\sum_{i=1}^{B}\left[-\mathbb{E}_{z\sim\text{vMF}(\boldsymbol{\mu}_i, \kappa_i)}\left\{\log\sum_{c=1}^{C}\right.\right.$$

$$\left.\left. \exp\left(\text{part1} - \text{part2}\right)\right\} + \beta \cdot (\exp_{p_y} \cdot \exp_z)_i \right], \tag{3}$$

$$\text{part1} = \log C_d\left(\|\boldsymbol{w}_c\|^2\right),$$

$$\text{part2} = \log C_d\left(\|\boldsymbol{w}_c\|^2 + \beta^2 + 2\beta \cdot \boldsymbol{z}^\top \boldsymbol{w}_c\right).$$

part1 - part2 describes the relative similarity between the embedding vector and the class center. For in-class samples, this difference should be maximized; for out-of-class samples, it should be minimized, thereby achieving discriminative constraints that cluster in-class samples and separate out-of-class samples. This loss guides the embedding features to form a discriminative distribution in spherical space by maximizing consistency between in-class samples and class centers while minimizing interference from out-of-class samples. The first term of the loss function achieves this by sampling.

## 4 Experiments

We conducted image retrieval experiments on three widely used benchmark datasets: MIRFLICKR-25K, NUS-WIDE, and MS COCO. By comparing our proposed algorithm with existing methods across multiple evaluation metrics, we systematically assessed its performance in hash-based retrieval tasks.

### 4.1 Datasets and Experimental Settings

#### 4.1.1 Datasets

MIRFLICKR-25K: After excluding unlabeled samples in the experiments, 24,581 valid samples were obtained. NUS-WIDE: For experiments, only samples from 21 high-frequency cat-

Table 1: Mean Average Precision (mAP) Performance Comparison Across Datasets

| Method | References | MIRFLICKR-25K@all | | | NUS-WIDE@5000 | | | MS COCO@all | | |
|---|---|---|---|---|---|---|---|---|---|---|
| | | 32bits | 64bits | 128bits | 32bits | 64bits | 128bits | 32bits | 64bits | 128bits |
| DPSH | IJCAI 2016 | 0.761 | 0.764 | 0.765 | 0.824 | 0.835 | 0.843 | 0.598 | 0.601 | 0.605 |
| DSH | CVPR 2016 | 0.712 | 0.716 | 0.723 | 0.787 | 0.797 | 0.811 | 0.584 | 0.585 | 0.595 |
| HashNet | AAAI 2016 | 0.707 | 0.712 | 0.706 | 0.757 | 0.768 | 0.775 | 0.542 | 0.553 | 0.563 |
| GreedyHash | NeurIPS 2018 | 0.704 | 0.726 | 0.750 | 0.799 | 0.830 | 0.846 | 0.611 | 0.649 | 0.670 |
| CSQ | CVPR 2020 | 0.727 | 0.740 | 0.748 | 0.821 | 0.830 | 0.833 | 0.586 | 0.630 | 0.644 |
| OrthoHash | NeurIPS 2021 | 0.715 | 0.734 | 0.744 | 0.805 | 0.817 | 0.828 | 0.601 | 0.640 | 0.660 |
| IDHN | TMM 2022 | 0.739 | 0.728 | 0.711 | 0.823 | 0.833 | 0.837 | 0.588 | 0.550 | 0.486 |
| HyP$^2$Loss | ACM MM 2022 | 0.731 | 0.742 | 0.749 | 0.829 | 0.838 | 0.844 | 0.626 | 0.640 | 0.660 |
| HSWD | CVPR 2022 | 0.720 | 0.745 | 0.763 | 0.719 | 0.810 | 0.842 | 0.546 | 0.584 | 0.611 |
| HHF | TMM 2023 | 0.592 | 0.647 | 0.604 | 0.815 | 0.803 | 0.777 | 0.608 | 0.607 | 0.581 |
| CenterHash | CVPR 2023 | 0.767 | 0.786 | 0.780 | 0.856 | 0.859 | 0.864 | 0.661 | 0.681 | 0.683 |
| CFBH | TMM 2024 | 0.743 | 0.745 | 0.754 | 0.851 | 0.859 | 0.862 | 0.630 | 0.644 | 0.655 |
| DCPH | NEUCOM 2025 | 0.757 | 0.785 | 0.802 | 0.788 | 0.807 | 0.836 | 0.673 | 0.691 | 0.711 |
| DCGMH | AAAI 2025 | 0.787 | 0.800 | 0.805 | 0.844 | 0.850 | 0.857 | 0.648 | 0.666 | 0.701 |
| vMFDH | vMFDH | 0.801 | 0.817 | 0.824 | 0.853 | 0.863 | 0.869 | 0.673 | 0.722 | 0.748 |

Table 2: Normalized Discounted Cumulative Gain at 1000 (NDCG@1000) Performance

| Method | References | MIRFLICKR-25K | | | NUS-WIDE | | | MS COCO | | |
|---|---|---|---|---|---|---|---|---|---|---|
| | | 32bits | 64bits | 128bits | 32bits | 64bits | 128bits | 32bits | 64bits | 128bits |
| DPSH | IJCAI 2016 | 0.415 | 0.425 | 0.438 | 0.380 | 0.404 | 0.427 | 0.266 | 0.276 | 0.304 |
| DSH | CVPR 2016 | 0.321 | 0.326 | 0.342 | 0.340 | 0.331 | 0.358 | 0.252 | 0.268 | 0.273 |
| HashNet | AAAI 2016 | 0.361 | 0.378 | 0.385 | 0.316 | 0.336 | 0.349 | 0.221 | 0.254 | 0.281 |
| GreedyHash | NeurIPS 2018 | 0.397 | 0.440 | 0.466 | 0.320 | 0.372 | 0.411 | 0.439 | 0.472 | 0.497 |
| CSQ | CVPR 2020 | 0.422 | 0.440 | 0.448 | 0.397 | 0.407 | 0.418 | 0.385 | 0.421 | 0.444 |
| OrthoHash | NeurIPS 2021 | 0.414 | 0.442 | 0.454 | 0.391 | 0.417 | 0.431 | 0.383 | 0.424 | 0.444 |
| IDHN | TMM 2022 | 0.407 | 0.424 | 0.433 | 0.388 | 0.425 | 0.439 | 0.325 | 0.319 | 0.267 |
| HyP$^2$Loss | ACM MM 2022 | 0.411 | 0.442 | 0.453 | 0.396 | 0.419 | 0.437 | 0.372 | 0.397 | 0.421 |
| HHF | TMM 2023 | 0.342 | 0.427 | 0.396 | 0.392 | 0.380 | 0.375 | 0.537 | 0.538 | 0.535 |
| CenterHash | CVPR 2023 | 0.472 | 0.498 | 0.510 | 0.437 | 0.446 | 0.463 | 0.361 | 0.523 | 0.452 |
| CFBH | TMM 2024 | 0.459 | 0.476 | 0.480 | 0.435 | 0.458 | 0.467 | 0.460 | 0.469 | 0.491 |
| DCPH | NEUCOM 2025 | 0.394 | 0.432 | 0.469 | 0.356 | 0.373 | 0.417 | 0.453 | 0.495 | 0.533 |
| DCGMH | AAAI 2025 | 0.460 | 0.454 | 0.461 | 0.395 | 0.392 | 0.413 | 0.423 | 0.525 | 0.432 |
| vMFDH | vMFDH | 0.494 | 0.511 | 0.531 | 0.453 | 0.471 | 0.491 | 0.509 | 0.538 | 0.563 |

egories were retained, resulting in a final dataset size of 195,834. MS-COCO: For practical use, after filtering, 132,218 samples across 80 categories were retained.

### 4.1.2 Experimental Settings

We employ a pre-trained ResNet50 as the base network to extract deep semantic features. Model training utilizes the Adam optimizer, with learning rates for both the backbone network and classifier set to $1e-5$. The learning rate for the temperature parameter is separately set to $1e-3$, and the weight decay coefficient is set to $1e-4$. A mini-batch size of 128 is employed during training. These metrics include: Mean Average Precision (mAP), Normalized Discounted Cumulative Gain (NDCG) for the top 1000 results, Precision-Recall Curve (PR curve), Top-N Precision Curve, and Precision Analysis Curve within a Hamming Distance radius 2 (P@H 2).

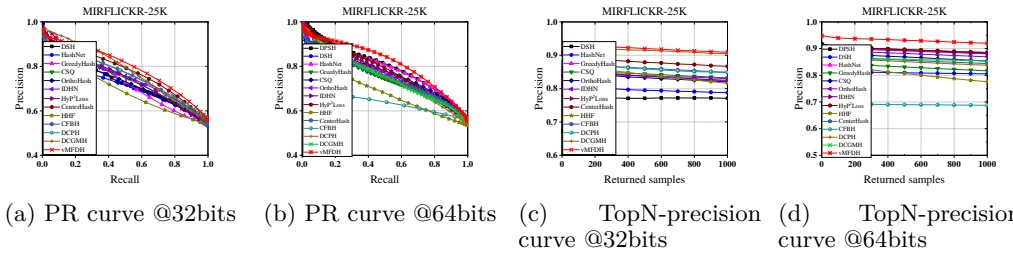

Figure 3: Performance curves on MIRFLICKR-25K dataset

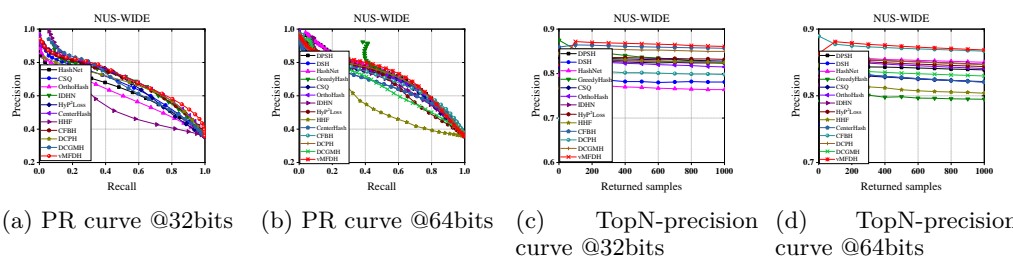

Figure 4: Performance curves on NUS-WIDE dataset

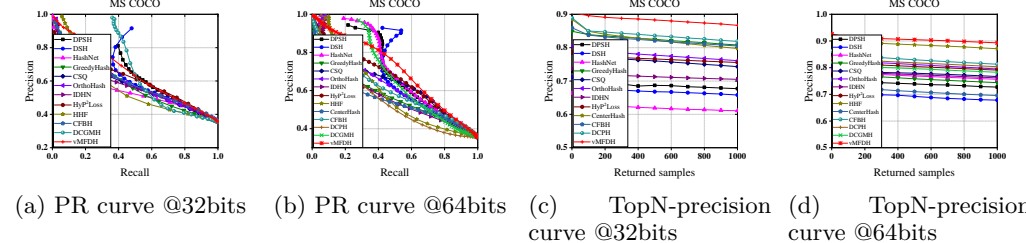

Figure 5: Performance curves on MS COCO dataset

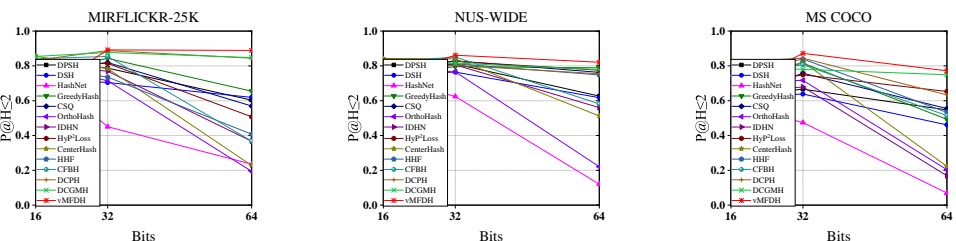

Figure 6: Precision at Hamming distance $\leq 2$ (P@H$\leq$ 2) across datasets

## 4.2 Comparison with the SOTA

To comprehensively evaluate the performance of vMFDH, we selected several representative deep hashing methods, including DSH Liu et al. (2016), HashNet Cao et al. (2017), GreedyHash Su et al. (2018), CSQ Yuan et al. (2020), OrthoHash Hoe et al. (2021), IDHN Zhang et al. (2020), HyP²Loss Xu et al. (2022), HWSD Doan et al. (2022), HHF Xu et al. (2023), CenterHash Wang et al. (2023c), CFBH Xiang et al. (2024), DCPH Qin et al. (2025), DCGMH Liu et al. (2025), and the traditional hashing method DPSH. All baseline methods employ a pre-trained ResNet50. We comprehensively evaluated vMFDH against baseline methods on MIRFLICKR-25K, NUS-WIDE and MS-COCO for 32–128-bit hash

Table 3: Ablation Study on mAP Performance Across Datasets

| Method | MIRFLICKR-25K | | | NUS-WIDE | | | MS COCO | | |
|--------|-------|-------|--------|-------|-------|--------|-------|-------|--------|
| | 32bits | 64bits | 128bits | 32bits | 64bits | 128bits | 32bits | 64bits | 128bits |
| vMFDH-L1 | 0.747 | 0.773 | 0.760 | 0.836 | 0.857 | 0.867 | 0.651 | 0.703 | 0.737 |
| vMFDH-L2 | 0.786 | 0.810 | 0.818 | 0.836 | 0.857 | 0.867 | 0.651 | 0.703 | 0.737 |
| vMFDH-S | 0.789 | 0.811 | 0.821 | 0.842 | 0.861 | 0.870 | 0.658 | 0.703 | 0.740 |
| vMFDH-N | 0.781 | 0.793 | 0.778 | 0.836 | 0.857 | 0.867 | 0.651 | 0.703 | 0.737 |
| DSH | 0.712 | 0.716 | 0.723 | 0.787 | 0.797 | 0.811 | 0.584 | 0.585 | 0.595 |
| vMFDH | 0.801 | 0.817 | 0.824 | 0.853 | 0.863 | 0.869 | 0.673 | 0.722 | 0.748 |

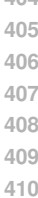
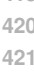

Figure 7: Top-10 retrieval images of our proposed vMFDH, DSH, OrthoHash and DCGMH on MS COCO dataset with 64-bit binary codes.Red boxes indicate incorrect retrieval results.

codes, using mAP and NDCG@1000 as metrics. Visual comparisons relied on PR curves, TopN-precision curves, and P@H 2 curves.

Table 1 2 presents mAP and NDCG@1000 values across different datasets and hash lengths. Our proposed model accurately preserves semantic similarity, outperforming the current state-of-the-art algorithm CenterHash across all datasets. It achieves mAP improvements ranging from1.07% - 8.41% at different bit lengths, validating vMF's exceptional performance in high-dimensional embedding modeling.

Figures 3 4 5 6 visualize comparisons. Benefiting from vMF hash's strengths in embedding modeling, semantic preservation, and hash code compactness, vMFDH achieves better retrieval performance: its PR curve outperforms most baselines with slower precision decay in high-recall ranges; TopN-precision curve shows it maintains higher precision/stability

as returned samples increase; P@H$\leq$2 curve reveals smaller performance drops (vs. baselines) when hash length grows (due to discrete space sparsification). These visual results complement quantitative metrics, fully validating vMFDH's effectiveness.

### 4.3 Ablation Study

We evaluated the contribution of each component in the vMFDH method through ablation experiments, designing four variants—vMFDH-L2, vMFDH-L1, vMFDH-S, and vMFDH-N—with DSH included as the baseline model. '-L1' replaces the vMF projection with L1 norm, constraining features to L1 space; '-L2' replaces the vMF projection with L2 norm, allowing features to freely distribute across spherical space; '-S' denotes forcing features to uniformly distribute on the sphere; '-N' removes all spatial constraints.

The experimental results in Table 3 show the mAP of vMFDH and its variants across different hash code lengths. On MIRFLICKR-25K with a 32-bit code, vMFDH-L1 achieves 0.747 and vMFDH-L2 achieves 0.786, both lower than vMFDH's 0.801. This highlights the benefit of spherical space constraints in vMFDH, where the vMF projection's intra-class feature clustering plays a key role. vMFDH-S, which enforces uniform spherical distribution, achieves an mAP of 0.740 at 128 bits on MS COCO, slightly below vMFDH's 0.748. This confirms that vMF projections effectively cluster similar features within spherical regions, aiding semantic similarity encoding. Without constraints, the mAP on MIRFLICKR-25K with a 64-bit code drops to 0.793, significantly lower than vMFDH's 0.817, validating the importance of spherical constraints and vMF distribution in model performance. Finally, on NUS-WIDE at 128 bits, DSH achieves 0.811, while vMFDH outperforms it with 0.869. This demonstrates that vMFDH's use of feature vector directional distributions, as opposed to Euclidean distances, results in more precise semantic similarity matching and more discriminative binary hash codes.

### 4.4 Top-10 retrieval results

To evaluate the image retrieval performance of the proposed algorithm, we compared the Top-10 retrieval results of the vMFDH framework with several mainstream deep hashing methods, including DSH, OrthoHash, and DCGMH, on the 64-bit discrete-coded MS COCO dataset. As shown in Figure 7, for the query images (bus, mobile phone, and giraffe), vMFDH consistently achieved the highest number of relevant samples in the top-10 results. In contrast, the retrieval results from DSH, OrthoHash, and DCGMH often displayed noticeable semantic mismatches with the query images. This improved performance of vMFDH can be attributed to its incorporation of the von Mises-Fisher (vMF) distribution, which enhances semantic similarity matching. Additionally, the use of a specialized loss function for regulating feature clustering enables vMFDH to generate compact and well-separated binary hash codes, improving retrieval accuracy.

## 5 Conclusion

This paper proposes a deep hashing framework based on the vMF distribution. By incorporating the von Mises-Fisher (vMF) distribution, this framework models the directional distribution of feature vectors, achieveing more precise semantic similarity matching. Leveraging the maximum entropy property of vMF distributions on hyperspheres, our method better accommodates the intrinsic distribution patterns of high-dimensional feature vectors, enhancing the model's generalization robustness. Furthermore, by designing a novel loss function to regulate the distribution concentration parameter,guiding the model to generate binary hash codes that balance intra-class compactness and inter-class separability. Extensive experiments demonstrate that our proposed vMFDH framework outperforms existing state-of-the-art deep hashing frameworks in retrieval performance.

Future work may integrate the vMF distribution into multimodal hashing scenarios to enhance multimodal retrieval performance.

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

# A  Appendix

## A.1  Sensitivity to the Parameters

We examined how parameters affect model performance through sensitivity experiments on the MIRFLICKR-25K, NUS-WIDE, and MS COCO datasets.These experiments primarily investigated the effects of the initial temperature $T$ and $\kappa$ confidence level on model performance, with results shown in Figure 8.In our proposed vMFDH framework, the temperature initialization $T$ and $\kappa$-confidence jointly regulate the initialization of the feature probability distribution and the modeling of the directional distribution of high-dimensional features.During experiments, we varied the temperature initialization $T = \{0, 0.2, 0.4, 0.6, 0.8\}$ and $\kappa$-confidence $= \{0.5, 0.6, 0.7, 0.8, 0.9\}$, evaluating model performance using mean average precision (mAP).When selecting $T = 0$ and $\kappa$-confidence $= 0.7$, our framework demonstrated superior and stable performance across all three datasets.

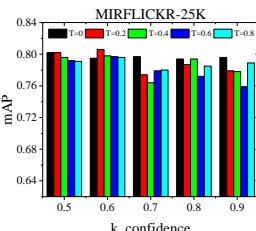 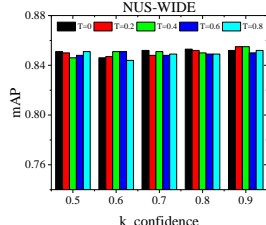 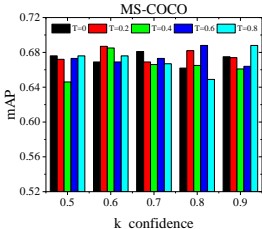

Figure 8: mAP results of vMFDH with different parameters $T$ and $\kappa$-confidence on MIRFLICKR-25K, NUS-WIDE, and MS COCO datasets (32bits)

## A.2  Training and Encoding Time

To evaluate the efficiency of our proposed framework,, we tested the training time and encoding time for 32-bit binary codes on the MIRFLICKR-25K dataset. For fair comparison, we employed a pre-trained ResNet50 architecture as the feature extractor. The experimental results are shown in Figure 9. The overall training time for deep hashing encompasses both feature representation learning and hash learning.

Although the training time of our proposed vMFDH framework is 1003.47 seconds, exceeding other traditional methods, the training process of the deep hashing framework is offline and does not affect retrieval performance. As a core metric for retrieval efficiency, the encoding time of the vMFDH framework is 16.91 seconds, lower than the high-performance CenterHash method, ensuring retrieval efficiency while improving performance.

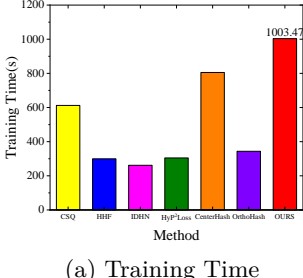

(a) Training Time

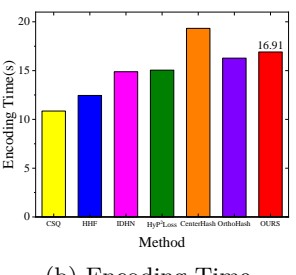

(b) Encoding Time

Figure 9: Comparison of training time and encoding time for deep hashing frameworks on MIRFLICKR-25K dataset (32bits)

### A.3    t-SNE results

To visualize compact discrete codes, we performed t-SNE visualization on the 64-bit binary codes obtained from our proposed vMFDH framework. Visualization comparisons were conducted alongside typical methods HashNet, OrthoHash, and CenterHash on the MS COCO dataset, with experimental results shown in Figure 10.The vMFDH framework achieves more precise semantic similarity matching by accurately modeling feature directions through the vMF distribution. Combined with the designed loss function, it dynamically learns hash codes, effectively enhancing the cohesion of similar samples and the separability of dissimilar samples. The figure reveals: Compared to HashNet, vMFDH exhibits clearer boundaries between clusters of different categories, avoiding the overlap between gray and orange samples. Relative to OrthoHash, vMFDH achieves a more compact distribution of similar samples, significantly reducing the dispersion between red and gray categories. Compared to CenterHash, vMFDH maintains category cohesion while further increasing the spatial distance between different categories.These results validate that the vMFDH framework can learn more discriminative binary codes, leading to superior retrieval performance.

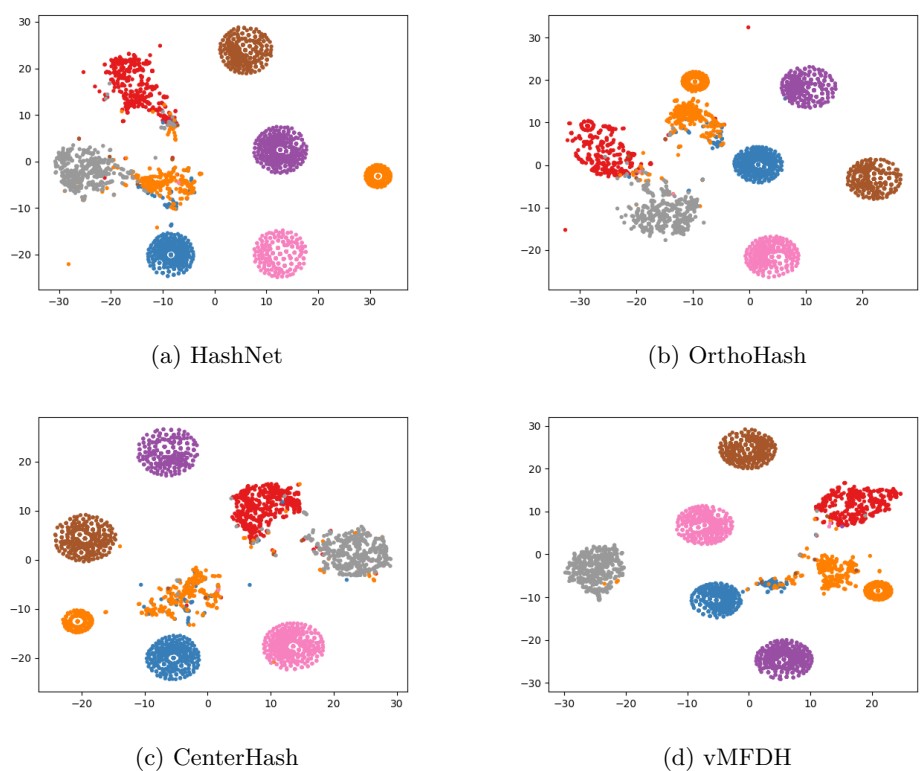

(a) HashNet

(b) OrthoHash

(c) CenterHash

(d) vMFDH

Figure 10: t-SNE visualization of 64-bit binary codes from the MS COCO set using HashNet, OrthoHash, CenterHash, and vMFDH. Different colors represent different categories.