# OpenReview forum: "Deep Hashing Based on the von Mises-Fisher Distribution for Fast Large-Scale Image Retrieval"
_ICLR.cc/2026/Conference — Submitted to ICLR 2026_

### Official Review · Reviewer_cey4 · 2025-10-28

**Soundness:** 2
**Presentation:** 3
**Contribution:** 2
**Rating:** 2
**Confidence:** 4

**Summary:**

This paper proposes vMF-DH, a  deep hashing method that addresses limitations of Euclidean-based and deterministic spherical hashing. By modeling features with von Mises-Fisher distributions on the unit hypersphere, it effectively captures directional similarity while enhancing noise robustness through maximum entropy properties. The designed vMF-Hash loss ensures compact intra-class and separable inter-class binary codes. Comprehensive experiments demonstrate state-of-the-art retrieval performance across multiple benchmarks, validating the method's effectiveness.

**Strengths:**

- The use of the von Mises-Fisher distribution is new to deep hashing. It leverages its maximum entropy property on the hypersphere for handling high-dimensional directional data.

- The authors present the experimental evaluation on several benchmark datasets including MIRFLICKR-25K, NUS-WIDE, MS COCO, using several evaluation metrics including mAP, NDCG@1000, P@H≤2. The proposed vMF-DH method shows superior performance across different bit lengths and datasets.

**Weaknesses:**

- The novelty of this work seems limited. The authors adapt a probabilistic vMF model for hash code learning without proper citation specifically in Section 3.2.
The paper does not adequately distinguish its contribution from prior work that also leverages cosine similarity or spherical embeddings (e.g., CSQ, CenterHash). The specific advantage of a probabilistic vMF model over these alternatives is not deeply discussed or empirically verified.

- The core methodology is described in a confusing and insufficiently detailed manner. It is required to detail technical parts.

- The deep analysis on empirical results why this work is better than the baselines is not given. The empirical studies should better verify the theoretical claims.

- The computational complexity is not discussed.

- The claim that vMF provides better "adaptability" and "robustness to noise" requires verification.

- The writing requires further improvement. The writing contains repetitive phrases and convoluted sentences, especially in the Methodology and Loss Function sections.

**Questions:**

Pls see the comments in Section Weaknesses.

---

### Official Review · Reviewer_KL9L · 2025-10-30

**Soundness:** 3
**Presentation:** 2
**Contribution:** 2
**Rating:** 2
**Confidence:** 4

**Summary:**

This paper studies deep hashing, and uses of the von Mises-Fisher distribution to model the directional similarity of high-dimensional features on a unit hypersphere. Experiments are conducted on three datasets and ResNet50 is used as backbone. Comparisons show boosted performance of the proposed method.

**Strengths:**

1. The proposed method works well as shown in experiments.
2. This paper is well motivated and reasonable in methodolgy.

**Weaknesses:**

1. The presentation should be improved. The figure 1 is not effective in showing the intuition of the proposed method. The loss function in eq. 3 is complicated. Fig.7 is informative enough, and should be moved to appendix.
2. The proposed method does not always show best performance in comparison with existing works. Some improvements are marginal in some settings.
3. The experiments are not quite convincing enough. Only one backbone is tested. ResNet50 is an outdated backbone. The study of hyperparameters should also be provided.
4. The complexity and computational cost of the proposed method should also be discussed and evaluated.
5. The proposed method relies on triplets as training samples, which could be sensitive to triplet sampling strategy, and leads to degraded training efficiency.

**Questions:**

1. The authors should conducted more extensive experiments by using more recent backbones.
2. The generalization capability of tuned hyperparameter across datasets should be evaluated.
3. The training efficiency should be discussed.

---

### Official Review · Reviewer_Y35N · 2025-11-01

**Soundness:** 3
**Presentation:** 2
**Contribution:** 3
**Rating:** 6
**Confidence:** 2

**Summary:**

This paper focuses on addressing limitations of existing deep hashing methods in large-scale image retrieval and proposes a novel solution called von Mises-Fisher Deep Hashing (vMF-DH). It points out that current methods either depend on Euclidean distance (ignoring directional similarity and causing retrieval errors in high-dimensional scenarios) or use deterministic spherical projections (neglecting feature probability distributions and being vulnerable to noise). To solve these issues, the paper introduces the von Mises-Fisher (vMF) distribution to map features onto the unit hypersphere, modeling directional distributions (cosine similarity) and leveraging the vMF distribution’s maximum entropy property to boost adaptability and robustness. It also designs the vMF-Hash loss function, which uses the vMF concentration parameter to regulate feature clustering, ensuring generated binary hash codes are compact within classes, well-separated across classes, and highly discriminative. Extensive experiments on multiple benchmark datasets verify that vMF-DH outperforms state-of-the-art deep hashing methods in retrieval accuracy and robustness.

**Strengths:**

The paper’s strengths lay a solid foundation for its weak acceptance: it astutely targets two core flaws in existing deep hashing methods—overdependence on Euclidean distance and disregard for feature probability distributions, addressing a pressing need in large-scale image retrieval.
Its integration of the von Mises-Fisher (vMF) distribution to model directional features and leverage its maximum entropy property is a theoretically innovative choice, while the custom vMF-Hash loss function effectively ensures hash codes are class-compact and inter-class separated, a key requirement for retrieval performance. Additionally, rigorous experiments across multiple benchmarks provide clear empirical proof that vMF-DH outperforms state-of-the-art methods, confirming its practical utility.

**Weaknesses:**

The paper lacks a systematic analysis of how the vMF concentration parameter’s magnitude influences model behavior, leaving a gap in understanding the method’s internal dynamics; it also does not compare vMF-DH’s memory footprint with baselines nor test its performance on datasets with highly imbalanced class distributions, which limits confidence in its generalizability and deployment potential.

**Questions:**

The paper uses the vMF concentration parameter in the vMF-Hash loss function to regulate feature clustering, but it does not elaborate on the selection strategy of this parameter. Could you explain how you determined the optimal range or initial value of the vMF concentration parameter in experiments, and whether there is an adaptive adjustment mechanism that can dynamically optimize this parameter according to different dataset characteristics?
The paper mentions that vMF-DH outperforms state-of-the-art methods in retrieval accuracy, but large-scale image retrieval also requires high computational efficiency. Can you provide specific experimental data (such as hash code generation time, query response time) to compare the computational efficiency of vMF-DH with baseline methods, and explain whether the introduction of the vMF distribution increases model complexity or inference latency?

---

### Official Review · Reviewer_G89N · 2025-11-01

**Soundness:** 1
**Presentation:** 1
**Contribution:** 2
**Rating:** 2
**Confidence:** 4

**Summary:**

The paper proposes a deep hashing method (vMF-DH) that maps features onto the unit hypersphere using the von Mises–Fisher (vMF) distribution, aiming to model directional similarity rather than Euclidean distance. A new loss function is introduced to control feature concentration and encourage compact within-class and separated across-class hash codes. Experiments on standard image retrieval benchmarks (MIRFLICKR-25K, NUS-WIDE, MS-COCO) are reported to show improvements over prior deep hashing methods.

**Strengths:**

1) Empirical results seem to show good results relative to prior work.
2) The idea of using the von Mises-Fisher Distribution to model unit normalized embedded data seems to be a good approach to take.

**Weaknesses:**

1) The paper is not written well at all, in many respects, as I will specify next. Therefore, I find it far from ICLR standards in its current form.
2) The main motivation of the paper is unclear - What exactly is the problem with prior approaches and how does the current one deal with them. The reasoning is based on sentences like "High-dimensional data inherently
exhibits spherical distribution characteristics" (line 121) and " the directional clustering properties of high-dimensional features" (line 157), which are not generally true. Neither is any empirical evidence provided.
3) The main Figures (1 and 2) are very confusing and do not assist in understanding the principles behind the method. For example, Figure 2 contains quite a few color inconsistencies, notations that appear nowhere and part that are not refered to.
4) Method description is incomplete and unclear, including unclear and missing notations. E.g. what is C_d(.)? What are exp_{p_y} and exp_z?
5) Results might be strong, but they are not presented in an understandable manner - Relevant dataset details a mostly missing, measures not explained, etc'.

**Questions:**

.

---

### Meta-Review · Area_Chair_2yEg · 2026-01-03

**Summary:**

This paper proposes a deep hashing method based on the von Mises–Fisher distribution for learning compact binary codes. The reviewer acknowledges the motivation and the experimental evaluation on multiple benchmark datasets, but raises concerns regarding the limited novelty, insufficient methodological clarity, lack of in-depth empirical analysis to support the claimed advantages, and missing discussion on computational complexity. In the absence of a rebuttal or discussion, these concerns remain central to the decision.

**Reviewer Concerns:**

No rebuttal or author response was provided, and there was no reviewer discussion. Therefore, the concerns raised by the reviewer remain unaddressed.

**Reviewer Scores:**

As no rebuttal or discussion took place, the reviewer would be expected to maintain the original score.

---

### Decision · Program_Chairs · 2026-01-26

Reject